# Gut Microbiota Manipulation as a Tool for Colorectal Cancer Management: Recent Advances in Its Use for Therapeutic Purposes

**DOI:** 10.3390/ijms21155389

**Published:** 2020-07-29

**Authors:** Federica Perillo, Chiara Amoroso, Francesco Strati, Maria Rita Giuffrè, Angélica Díaz-Basabe, Georgia Lattanzi, Federica Facciotti

**Affiliations:** 1Department of Experimental Oncology, IEO European Institute of Oncology IRCCS, 20139 Milan, Italy; federica.perillo@ieo.it (F.P.); chiara.amoroso@ieo.it (C.A.); MariaRita.giuffre@ieo.it (M.R.G.); angelicajulieth.diazbasabe@ieo.it (A.D.-B.); georgia.lattanzi@ieo.it (G.L.); 2Department of Oncology and Hemato-Oncology, Università degli Studi di Milano, 20135 Milan, Italy

**Keywords:** colorectal cancer, gut microbiome, live biotherapeutic products

## Abstract

Colorectal cancer (CRC) is a multifaceted disease influenced by both environmental and genetic factors. A large body of literature has demonstrated the role of gut microbes in promoting inflammatory responses, creating a suitable microenvironment for the development of skewed interactions between the host and the gut microbiota and cancer initiation. Even if surgery is the primary therapeutic strategy, patients with advanced disease or cancer recurrence after surgery remain difficult to cure. Therefore, the gut microbiota has been proposed as a novel therapeutic target in light of recent promising data in which it seems to modulate the response to cancer immunotherapy. The use of microbe-targeted therapies, including antibiotics, prebiotics, live biotherapeutics, and fecal microbiota transplantation, is therefore considered to support current therapies in CRC management. In this review, we will discuss the importance of host−microbe interactions in CRC and how promoting homeostatic immune responses through microbe-targeted therapies may be useful in preventing/treating CRC development.

## 1. Background

Colorectal cancer (CRC) is the third most commonly diagnosed malignancy and the fourth leading cause of cancer death in the world [1]. CRC is a heterogeneous disease with a wide range of long-term outcomes and responses to treatment. Genetic and environmental factors contribute to the etiology and progression of the disease. Several risk factors have been identified, including positive family history, smoking, alcohol intake, lifestyle, and cultural and social practices [2]. Moreover, emerging evidence suggests that diet has an important impact on the risk of CRC development [3]. Over the past three decades, molecular genetic studies have revealed some critical mutations underlying the pathogenesis of the sporadic and inherited forms of CRC [4], including mutational inactivation of the adenomatous polyposis coli (APC) tumor suppressor [5], resulting in overactivation of the Wnt/β-catenin signaling pathway, dysregulated cell proliferation and adenoma development [6], or microsatellite instability, assessed with the detection of mono- and di-nucleotide tracts selected by the National Cancer Institute consensus conference [4,7].

Current treatments for CRC include endoscopic and surgical local excision, downstaging preoperative radiotherapy and systemic therapy, extensive surgery for locoregional and metastatic disease, local ablative therapies for metastases, palliative chemotherapy, targeted therapy, and immunotherapy [8]. Although these treatments have doubled the overall survival of patients up to 3 years for advanced stages of the disease, CRC remains associated with poor prognosis and very low rates of long-term survival [9]. The development and progression of CRC are multi-factorial processes which are associated also with the progressive failure of immunosurveillance, which is the natural and/or therapy-stimulated capacity of the immune system to control cancer progression [10]. However, the role of altered mucosal immune surveillance in the development of CRC has not yet been fully understood [11].

The role of the gut microbiota in cancer biology has been increasingly recognized as an environmental factor favoring CRC development. Indeed, the gut microenvironment harbors a complex microbial ecosystem comprising approximately 3 × 10^13^ bacteria and other microorganisms such as fungi, phages, archaea, and protists [12], which are confined into the intestinal lumen by the epithelial cell lining, the mucus layers, and the production of antibacterial peptides and bioactive molecules [13]. The mutual interaction between the gut microbiota and the host is further highlighted by its role in inducing immune maturation [14]. For this reason, a CRC-associated microbial dysbiosis can alter the delicate equilibrium between the gut microbiota and the host’s immune system, contributing to cancer initiation and/or progression [15]. The spatial organization of multispecies bacterial communities in higher-order structures, termed biofilms, appears to be indispensable for CRC initiation. According to the adenoma–carcinoma sequence model proposed by Fearon and Vogelstein [16], microbial biofilm may be regarded as an independent “driver” at an early stage of CRC carcinogenesis, before the malignant transformation from adenoma to carcinoma. Here, we will recapitulate how the interconnection between the intestinal microenvironment, CRC onset, and the gut microbiota composition may influence therapies’ outcomes. We will discuss the importance of gut microbiota modulation in CRC patients through dietary interventions as well as microbiome biomodulators, including anti-, pro-, pre-, and post-biotics and fecal microbiota transplantation, and how these therapeutic strategies can help in preventing/treating CRC.

## 2. Role of Gut Microbiota in CRC

### 2.1. Pro-Tumorigenic Roles of the Gut Microbiota in CRC

The gut microbiota has been linked to carcinogenesis and colorectal tumor progression [15]. Pathobionts (microorganisms commonly living in the gut that become harmful under certain circumstances) colonizing the intestine may produce genotoxins, metabolites that induce DNA damage, which can lead to alterations in the tumor microenvironment (TME), inducing subsequent changes in the abundance of colonic intrinsic pathogenic members (Table 1) [17]. For instance, *Escherichia coli* is a bacterium commonly found in the human gut. However, some strains are pathogenic and can promote disease via several virulence factors [18]. The *E. coli*-derived colibactin toxin, encoded by the pathogenicity island *pks*, is frequently associated with human colorectal carcinogenesis, as demonstrated in human and animal models in which *pks*^+^
*E. coli* strains induce double-strand DNA breaks, mutations, chromosomal rearrangements, and cell cycle arrest [19]. Accordingly, *pks*^+^
*E. coli* strains induce single-base substitution/indels mutational patterns, which are predominantly detected in CRC patients [20]. In addition, the enterotoxigenic *B. fragilis* (ETBF), an anaerobic bacterium found in the human intestinal microbiota, produces an enterotoxin (*B. fragilis* toxin, BFT), which is highly associated with CRC [21], by activating β-catenin signaling as well as the secretion of interleukin (IL)-8 in colonic epithelial cells, leading to persistent cellular proliferation [22]. The alteration of the gut microbiota composition, known as “dysbiosis”, has been observed in CRC patients [23]. A large body of literature has reported that fecal and intestinal mucosa samples from CRC patients display a lower bacterial diversity compared to healthy individuals [24,25]. In addition, CRC patients show significant alterations in specific bacterial taxa, with a potentially detrimental impact on mucosal immune responses [25]. In particular, the CRC-associated microbiota is characterized by the increased abundance of *Enterobacteriaceae*, *Streptococcus*, and typical genera belonging to the oral microbiota such as *Fusobacterium*, *Gemella*, *Peptostreptococcus*, *Prevotella*, *Solobacterium*, and *Parvimonas* [26,27,28]. On the contrary, the *Firmicutes* phylum (especially the *Ruminococcaceae* and *Lachnospiraceae* families) as well as *Bifidobacterium, Odoribacter*, and *Streptococcus* are substantially underrepresented in CRC patients [29,30]. In fact, gut microbiota transplant from CRC patients into mice is sufficient to disrupt the intestinal barrier, leading to low-grade inflammation and dysbiosis. Indeed, conventional and germ-free (GF) mice transplanted with stool samples from CRC patients developed high-grade dysplasia and macroscopic polyps concomitantly with a higher proportion of colonic Ki-67 positive proliferating cells as well as increased expression of C-X-C motif chemokine receptor 1, C-X-C motif chemokine receptor 2, IL-17A, IL-22, and IL-23A cytokines. [31]. Similarly, Apc^Min/+^ mice (carrying a mutation predisposing them to intestinal adenoma and tumor formation) transplanted with feces from CRC patients showed an increase in the number of intestinal tumors, downregulated expression of mucin-2, regenerating islet-derived protein and intestinal secretory immunoglobulin A, upregulation of the NLRP3 inflammasome, and increased production of pro-inflammatory cytokines IL-1β and tumor necrosis factor (TNF)-α. The hyperproliferative and pro-inflammatory phenotype of the CRC associated-mucosa can be attributed to a dysbiotic microbiota, defined as an “imbalanced” gut microbial composition, associated with disease, that promotes the activation of the Wnt signaling pathway [32]. Indeed, alterations of the gut microbiota can also influence tumor development and progression by impairing immunosurveillance [33], leading to features of exhaustion once tumors are established [34], together with a pro-inflammatory phenotype at early stages [23,35,36,37,38]. *Fusobacterium nucleatum* (*Fn*) is commonly found in colorectal tissue with high-grade dysplasia as well as in adenomas. It has been shown that *Fn* modulates several immune responses towards CRC tumor progression, influencing the pre-tumoral environment [39], inducing the production of inflammatory mediators (IL-6, IL-8, IL-1β) [40], transforming growth factor (TGF)-β and TNF-α [41], and suppressing antitumor immunity by the secretion of Fap2, a natural killer inhibitory ligand [42]. Similarly to colibactin and BFT, *Fn* mediates DNA damage and promotes tumor cell proliferation through the Wnt/β-catenin pathway [43]. Furthermore, bacterial metabolism might affect CRC progression by the production of oxidative stress molecules [30,44,45], promoting chronic inflammation and disrupting intestinal barrier integrity [36,46]. Moreover, choline degradation by gut bacteria contributes to colon cancer via the synthesis of trimethylamine N-oxide, a potential carcinogen [28].

### 2.2. Anti-Oncogenic Effects of the Gut Microbiota in Colon Cancer

On the other hand, many microbes within the gut microbiota have shown anticancer activities (Table 1). Some lines of evidence have shown that *Lactobacillus, Bifidobacterium*, and non-enterotoxigenic *Bacteroides fragilis* (NTBF) can control DNA damage by promoting epithelium renewal through lactic acid production, the modulation of the immune system by reducing the Th17 pool, enhancing major histocompatibility complex-II expression on dendritic cells, and improving natural killer cell and cytotoxic T cell recruitment and cytotoxicity [10,47,48,49,50,51]. Furthermore, the anticancer activities of the gut microbiota are also promoted by short chain fatty acid (SCFA) production. Indeed, one of the most prominent signatures of a healthy microbiota is the presence of SCFA-producing bacteria, such as members of the *Lachnospiraceae* and *Ruminococcaceae* families, among others [10,52]. SCFAs, by modulating histone deacetylase inhibitory activity, promote the accumulation and differentiation of Treg cells [53,54,55] controlling tumor progression. Accordingly, two anti-tumorigenic strains of the microbiota recently identified, *Faecalibaculum rodentium* and its human homologue, *Holdemanella biformis,* were able to control protein acetylation and tumor cell proliferation through the production of SCFAs, inhibiting the calcineurin and cytoplasmic 3 (NFATc3) activation pathway in mouse and human settings [56].
ijms-21-05389-t001_Table 1Table 1Summary of bacteria known to be involved in colorectal cancer progression and prevention.Name(Potential) Role in CRC OncogenicityMechanism of ActionReferencesProteobacteria, especially the *Enterobacteriaceae* familyPro-oncogenicOpportunistic pathogens, promotion of inflammation[22,37]*Escherichia coli*Pro-oncogenicDNA damage by colibactin, induction of a pro-inflammatory environment[18,19]Enterotoxigenic *Bacteroides fragilis* (ETBF)Pro-oncogenicColon cell hyperproliferation by β-catenin pathway activation and IL-8 production[20,21,50,57,58,59]*Fusobacterium nucleatum*Pro-oncogenicPromotion of inflammation, impairment of antitumor immunity, activation of β-catenin pathway, DNA damage[39,40,41,42]*Ruminococcaceae* familyAnti-oncogenicSCFA production[9,51]*Lachnospiraceae* familyAnti-oncogenicSCFA production[9,51]*Bifidobacteria*Anti-oncogenicSCFA production, reduction of pro-inflammatory cytokines, epithelial cell renewal[9,45,46,47]*Lactobacilli,* including *L. casei, L. plantarum, L. rhamnosus, and L. acidophilus*Anti-oncogenicSCFA production, reduction of pro-inflammatory cytokines, enhancement of antitumor immunity, epithelial cell renewal[46,47,50]Non-enterotoxigenic *Bacteroides fragilis* (NTBF)Anti-oncogenicBoost of antitumor immunity, amelioration of inflammation by PSA production[48,50]*Faecalibaculum rodentium* and *Holdemanella biformis*Anti-oncogenicSCFA production[55]*Akkermansia muciniphila*Anti-oncogenicSCFA production, regulation of intestinal barrier integrity[9,60,61]*Enterococcus faecalis*Anti-oncogenicImprovement of intestinal inflammation[62]

## 3. Lifestyle, Diet, and Microbiota in the Early Onset of CRC

Early-onset colorectal cancer (EOCRC) is the second most common cancer and the third leading cause of cancer mortality in people <50 years of age in the USA. The incidence of EOCRC has been on the rise over the past four decades and it is expected to increase by >140% by 2030. At present, different established cancer drivers have been linked to EOCRC including diet, sedentary lifestyle, smoking, and alcohol [63]. The gut microbiota is probably at the intersection of these risk factors and EOCRC. Indeed, the gut microbiome and inflammation are key players and master regulators of CRC onset and progression, as discussed above. The World Cancer Research Foundation (London, UK) and the American Institute for Cancer Research (Washington, DC, USA) consider diet to be one of the most important exogenous factors in CRC etiology [64]. The use of dietary modifications to supplement conventional cancer therapy is, therefore, a practical approach that is receiving growing attention. Dietary composition also dictates nutrient availability in the TME. Manipulation of the metabolic environment of cancer cells markedly changes their metabolic activity, producing shifts in drug sensitivity, proliferation rate, and metabolic requirements. The diet also influences the composition of the gut microbiota and, thus, the effect that gut microbes exert on the above-mentioned mechanisms.

It is now widely recognized that the adoption of a westernized diet rich in red meat and saturated fat and low in fiber exerts a negative effect on intestinal homeostasis [65], also promoting gut dysbiosis and inflammation [50,66,67,68,69,70]. The putative indication of colon cancer risk and a high-fat, low-fiber western diet was evaluated in a two-week diet exchange study among rural Africans, who usually have a low-fat, high-fiber diet, and African Americans, who usually consume the western diet. The diet exchange resulted in remarkable reciprocal changes in microbiota composition, as demonstrated by a shift in African Americans fed with the high fiber diet from *Bacteroides* and butyrate-producing bacteria (e.g., *Roseburia intestinalis* and *Clostridium symbiosum*) towards stronger co-occurrence patterns, including *Firmicutes*, which are typically associated with complex carbohydrate fermentation. On the other hand, the low-fiber/high-fat intervention was associated with an increase in *F. nucleatum*. Moreover, the latter were characterized by an increase in proliferative and inflammatory markers such as Ki67 and cluster of differentiation (CD)3^+^ intraepithelial lymphocytes and CD68^+^ lamina propria macrophages [71]. It has been postulated that the higher risk of developing CRC through a high-fat diet (HFD) may be due to an increase in the intestinal secretion of primary bile acids, converted into secondary bile acids by the gut microbiota, such as deoxycholic acid and lithocholic acid, which have been associated with a great increase in intestinal tumor formation and inflammatory damage in mice [72].

Changes not only in the diet but also in the use of food additives (used to extend the shelf-life of processed foods) have resulted in a considerable shift in food quality and increased risk of CRC onset. It is well known that nitrite and nitrate consumption, rich in processed meats, can lead to the formation of N-nitroso compounds by gut microbes, some of which are carcinogenic [73]. The health and regulatory issues related to the addition of food ingredients are too vast to cover in this review and have been covered elsewhere [74]; nevertheless, it is worthwhile to consider a few examples of food additives that modulate the gut microbiome and the host inflammatory status, factors associated with CRC development [73]. Monosodium glutamate is an additive used to enhance the flavor of savory foods able to induce obesity and diabetes. Interestingly, monosodium glutamate increases the susceptibility to CRC in models of inflammation-induced colorectal carcinogenesis [75]. Additionally, the food additive titanium dioxide, commonly used as a whitening and brightening agent, promotes colon inflammation and neoplastic lesions in chemically-induced carcinogenesis models [76,77]. Relatively low concentrations of two commonly used emulsifiers, carboxymethylcellulose and polysorbate-80, altered the gut microbiota composition and promoted low-grade intestinal inflammation in animal models [78,79]. These data suggest a role of food additives in the incidence of CRC development in humans. Altogether, these studies support a mechanistic link between the gut microbiota and CRC, since external factors that modulate the gut microbiome include not only stress and dietary factors but also elements previously thought to be disconnected from colon health, such as birth mode, breastfeeding behaviors, and maternal stress and nutrition [80,81,82].

## 4. Relevance of the Gut Microbiota in the Efficiency of Cancer Therapies

Advancements in CRC pathophysiological understanding have increased the array of treatment options for local and advanced disease, leading to individual therapeutic plans. Surgery is the cornerstone of curative treatment for patients with non-metastasized CRC [83]. In more advanced cases, neoadjuvant treatments, including preoperative chemotherapy, chemoradiotherapy, or radiotherapy, can reduce tumor load and stage and might be necessary to optimize the chances of a successful resection [84]. Current chemotherapies include both single-agent therapy, which is mainly fluoropyrimidine (5-FU)-based, and multiple agent regimens containing oxaliplatin (OX), irinotecan (IRI) and capecitabine (CAP or XELODA or XEL) [85]. Radical surgery and various chemotherapeutic agents can perceptibly create a state of dysbiosis, further exaggerating the influence of deleterious bacteria, reducing efficacy, and exacerbating the toxicity of chemotherapy [86]. In particular, compared with preoperative samples, fecal samples collected postoperatively exhibit a significant decrease in obligate anaerobes, tumor-related bacteria, and butyric acid-producing bacteria. However, a relevant increase in some conditional pathogens, such as *Bilophila*, *Eggerthella*, and *Anaerostipes*, was observed [87]. Moreover, chemotherapy also alters the intestinal microbiota through the so called “rebound effect”, which is characterized by a dramatic increase in pathogens and a shift in lactate-utilizing bacteria from *Veillonella* to *Butyricimonas* and *Butyricicoccus*, as well as a decrease in commensals [87]. Accordingly, stool samples from resected stage III CRC patients characterized by CapeOX chemotherapy-induced diarrhea presented lower bacterial community richness and diversity, with *Klebsiella pneumoniae* being the most predominant microbial species [88]. Specific members of the gut microbiota have been found to play a vital role in chemoresistance to 5-FU and OX therapy by mediating autophagy [89]. Indeed, the potential relationship between *Fn* infection and the chemotherapeutic efficacy of 5-FU was investigated both in vitro and in vivo. *F. nucleatum* load reduced the chemosensitivity of CRC cells to 5-FU by targeting MYD88 innate immune signaling and specific microRNAs responsible for the activation of the autophagy pathway. All these results demonstrate how *Fn* abundance is well correlated with a lower response in advanced CRC patients to 5-FU-based adjuvant chemotherapy after radical surgery [89,90]. In addition, other gut microbes might aggravate chemotherapy-related adverse reactions via the microbial metabolism of chemotherapy agents [91,92]. For instance, IRI effectiveness is severely limited by gastrointestinal tract toxicity caused by gut bacterial β-glucuronidase enzymes [91,93]. 

Besides these canonical treatments, other therapies have been explored for CRC management. Targeted therapies include four main groups of drugs: monoclonal antibodies against epidermal growth factor receptor (EGFR) (cetuximab and panitumumab), monoclonal antibodies against vascular endothelial growth factor (VEGF)-A (bevacizumab), fusion proteins that target multiple proangiogenic growth factors (e.g., aflibercept), and small-molecule-based multikinase inhibitors (e.g., regorafenib) [84]. These treatments can work on cancerous cells by directly inhibiting cell proliferation, differentiation, and migration, but they can also alter the TME to slow down tumor growth and promote a stronger immunosurveillance response [85]. Immune escape has been frequently identified in various cancers, including CRC [94]. One major explanation is tumor-related T cell inactivation and exhaustion via activation of co-inhibitory receptors, the so-called immune checkpoint receptors, on the surface of T cells [60], which include programmed cell death protein (PD)-1 and cytotoxic T lymphocyte antigen 4 (CTLA-4). Checkpoint inhibitors are now a standard of care in microsatellite-instable CRC patients [61,95,96]. The gut microbiota is an important player affecting the efficacy of the immune checkpoint blockade [97]. Initial findings by Vetizou et al. showed that the CTLA-4-targeting antibody ipilimumab could treat specific-pathogen-free mice but not GF mice. In addition, antibiotics including ampicillin, colistin, and streptomycin compromise the antitumor effects of this antibody, indicating the key role of the gut microbiota in immunotherapy outcomes [98]. Ipilimumab induces significant changes in the microbiome—in particular, a decrease in the bacterial orders *Bacteroidales* and *Burkholderiales*. Similarly, the presence of *Bifidobacterium* was positively correlated with anti-tumor T-cell responses in melanoma and bladder cancer mouse models treated with an anti-PD-L1 agent [99]. Gut microbes also affect the capacity of cytotoxic T cells to infiltrate the TME. Gut colonization with different *Bacteroidales* and non-*Bacteroidales* strains enhanced the efficacy of PD-1 and CTLA-4 monoclonal antibody therapies in GF mice due to the stronger immune-protective infiltration of CD8^+^ T cells. Unfortunately, the adverse effects of immunotherapy are dominated by autoimmune complications, such as fatal forms of colitis, as seen in metastatic melanoma of Ipilimumab-treated patients, which developed intestinal inflammation within the first 16 weeks of treatment [100].

## 5. Clinical Applications of the Gut Microbiota Modulation for CRC Prevention and Management

Several approaches, among which dietary interventions, antibiotic treatments, probiotics, prebiotics, and postbiotics, as well as fecal microbiota transplantation (FMT), have been explored to target and modulate gut microbiota composition, including both microbial physiology and/or their metabolites that cause or contribute to CRC directly or indirectly (Figure 1). Various experimental studies have deepened the understanding of the role of gut biomodulators and microbe-based treatments as antineoplastic agents, although a practical clinical application in CRC prevention and management is still largely lacking.

### 5.1. Dietary Interventions

As mentioned above, diet plays a significant role in shaping the microbiome and, therefore, in the management of CRC. While the impact of a westernized high-fat diet on CRC and on the gut microbiota has been well characterized, the protective effects of grain diets, known to be associated with low CRC risk, remain uncertain at the microbiota level. Yang et al. assessed the capacity of seven different grains to reduce CRC risk in mice fed with an HFD, showing that the consumption of non-glutinous rice, glutinous rice, and sorghum led to the highest reduction in CRC risk. In particular, non-glutinous rice stabilized key altered genera associated with CRC, including *Bacteroides*, *Lactobacillus*, *Ruminococcus*, and *Acinetobacter* [101]. Moreover, through a prospective cohort study, a diet rich in whole grains and dietary fiber was associated with a lower risk of developing *F. nucleatum*-positive CRC but not *F. nucleatum*-negative CRC, supporting a potential role of intestinal microbiota in the development of CRC [102]. Vitamin D supplementation reduces cancer incidence in mouse models of bacteria-driven colitis and CRC [57]. Azoxymethane/dextran sodium sulfate (AOM/DSS)-induced CRC mice fed with high doses of vitamin D showed not only improved body weight gain and less colon shortening but also a lower expression of inflammatory cytokines such as IL-6 and TNF-α. Vitamin D has also a significant regulatory effect on the homeostasis of the microbiota, especially on the regulation of the intestinal barrier integrity mediated by *Akkermansia muciniphila*, a mucin-utilizing bacterium [58,59].

Since there are no clear guidelines on the type of nutrition that could have a major impact on cancer incidence, various forms of reduced caloric intake, such as fasting, demonstrate a wide range of beneficial effects in preventing malignancies and increasing the efficacy of cancer therapies [103]. One of the main mechanisms through which fasting induces metabolic improvements is certainly mediated by the gut microbiota. For instance, every-other-day fasting led to an alteration of the gut microbiota composition, elevating fermentation products like acetate and lactate. Moreover, this dietary regimen enriched the levels of *Firmicutes* and also the production of SCFAs, decreasing *Bacteroidetes*, *Actinobacteria*, and *Tenericutes* [104]. In particular, food withdrawal decreased the abundance of potentially pathogenic *Proteobacteria* while elevating *A. muciniphila* in mice fed with HFD [105]. Due to various deficiencies in one or both key mitochondrial enzymes, tumors are not able to metabolize ketone bodies as an energetic source. Thus, the administration of a ketogenic diet (KD) may be a reasonable therapeutic strategy to inhibit tumor growth [106]. KDs are low-carbohydrate, high-fat diets, mimicking the metabolic state of fasting by inducing a physiological rise in acetoacetate and beta-hydroxybutyrate [107]. For cancer prevention, a high intake of mono-unsaturated fatty acids and n-3 polyunsaturated fatty acids could be hypothesized to be beneficial for promoting gut health [108], as demonstrated by the delayed tumor growth induced by a KD rich in omega-3 fatty acids in a CRC mouse xenograft model [106]. Lastly, supplementation with α-ketoglutarate, an important intermediary in the nuclear factor kappa light chain enhancer of activated B cells (NF-κB)-mediated inflammatory pathway, offered significant protection against CRC development in mice. Thus, α-ketoglutarate not only exhibited immunomodulatory effects mediated via the downregulation of IL-6, IL-22, TNF-α, and IL-1β cytokines but also minimized the frequency of opportunistic pathogens (*Escherichia* and *Enterococcus*), while it increased the populations of *Akkermansia*, *Butyricicoccus*, *Clostridium*, and *Ruminococcus*, suggesting that dietary α-ketoglutarate intervention may protect against inflammation-related CRC [109]. 

### 5.2. Antibiotics

Modulation of the gut microbiota through the use of antibiotics was partially evaluated in CRC, with only few studies present in the literature [110]. Cefoxitin, a semi-synthetic and broad-spectrum cephalosporin, induced a complete and durable clearance of enterotoxigenic *B. fragilis* colonization in previously ETBF-inoculated mice, with a concomitant decrease in median adenoma formation [111]. Consistent with the pro-tumorigenic Th17 immune response of ETBF [112,113], its eradication was accompanied by an abrupt reduction in colonic IL-17A levels, suggesting that other microbes are implicated in the IL-17 response [111]. Erythromycin has the ability to suppress the transcriptional activity of NF-κB and the activator protein-1 (AP-1), as well as its downstream targets, IL-6 and cyclooxygenase-2 (COX-2), in human CRC cells. Moreover, a reduction in *Il*-6 and *cox-2* mRNA expression was also observed in Apc^Min/+^ mice, in which the number of intestinal polyps was reduced as well [114]. Berberine (BBR), an isoquinoline molecule with antibacterial activity [115], has been used to treat *F. nucleatum* colonization in Apc^Min/+^ mouse models. BBR not only was able to reverse the microbiota imbalance induced by *Fn* but also blocked the secretion of mucosal immune factors, such as IL-21, IL-22, IL-31, and CD40L. In addition, this compound inhibited the *Fn*-induced activation of the Janus kinase/signal transducer and activator of transcription (JAK/STAT) and mitogen-activated protein kinase/extracellular signal-regulated kinases (MAPK/ERK) pathway [116]. Moreover, the microbial structure alteration, characterized by the increase in *Tenericutes* and *Verrucomicrobia*, was dramatically reversed in *Fn*-infected mice after BBR intervention, suggesting an antimicrobial intervention as a potential treatment for patients with *Fn*-associated CRC [117]. Metronidazole has also been explored as an alternative to treat *F. nucleatum* colonization. This antibiotic reduced the *Fn* load, cancer cell proliferation, and overall tumor growth in mice bearing colon cancer xenografts [117]. However, antibiotic administration, being the most aggressive means of manipulating gut microbiota composition, has been controversial in its role in cancer management. Although gut microbiome depletion was shown to inhibit cancer progression, accumulating lines of evidence highlight how antibiotics can compromise immunotherapy efficacy or induce disease progression by creating further microbial dysbiosis [118,119].

### 5.3. Probiotics

For CRC prevention and management, another potential strategy is represented by probiotics. Probiotics are living microorganisms which can confer positive effects on health by impacting on the resident microbiota, intestinal epithelium cells, and, globally, the immune system [120]. Nowadays, several bacterial species are used as probiotics, which are commercially available (Table 2).

Among them, lactic acid bacteria are the most frequently used, not only for their ability to contribute to colonization resistance [121] but also for their immunomodulatory effects [122]. Specific bacterial strains are able to prevent tumor development through the modulation of the immune system in CRC murine models. Oral treatment with *Lactobacillus casei* BL23, a probiotic strain well known for its anti-inflammatory [123] and anticancer properties [124], significantly protected mice against CRC development. In addition, this probiotic showed not only immunomodulatory effects by downregulating colonic IL-22 but also an antiproliferative effect mediated by the upregulation of caspase (casp)-7, casp-9, and Bik, as well as a decrease in Ki67 expression [125]. Probiotics have also been recently exploited to counteract chemotherapy-dependent dysbiosis, mucositis (inflammatory lesions of the oral and/or gastrointestinal tract caused by high-dose cancer therapies), post-surgical microbiota intestinal alterations, and relapses in CRC patients. Preclinical studies revealed that *L. rhamnosus* (Lcr35) reduces the severity of diarrhea and intestinal mucositis caused by adjuvant 5-FU-based chemotherapy in mice injected with CT26 colorectal adenocarcinoma cells. Moreover, Lcr35 treatment normalized the increased number of BCL2-associated X protein apoptotic and NF-κB-activated cells as well as the upregulated expression of TNF-α and IL-6 [126]. A recent randomized, double-blind, placebo-controlled trial (NCT03782428) revealed that treatment with six viable microorganisms of *Lactobacillus* and *Bifidobacterium* strains significantly reduced the levels of pro-inflammatory cytokines such as TNF-α, IL-6, IL-10, IL-12, IL-17A, IL-17C, and IL-22 and prevented post-surgical complications as well [127]. Probiotics are generally considered safe and well-tolerated for healthy subjects, but in patients with underlying medical conditions, their safety profile is uncertain [62]. Probiotic translocation, which refers to the entry of viable bacteria into extraintestinal sites, leads to the ensuing systemic or localized infections. Indeed, various case reports of probiotic-associated bacteremia, endocarditis, liver abscess, and pneumonia have been published [128]. Nevertheless, another theoretical risk regarding long-term probiotic use is the possible transmission of antibiotic-resistant genes via horizontal gene transfer [62].

### 5.4. Prebiotics

Prebiotics are non-digestible dietary compounds that stimulate the growth and activity of probiotics, conferring a competitive advantage to commensal bacteria capable of metabolizing these substrates or by increasing the production of beneficial metabolic products, such SCFAs, that result from their fermentation [129,130]. The health benefits of prebiotics goes beyond nutrition and they are gaining popularity among people (Table 3). However, great care must be taken to ensure their therapeutic efficacy, especially regarding intestinal tumors. The chemopreventive effect of galacto-oligosaccharides derived from lactulose revealed a remarkable reduction in colonic tumor numbers in a CRC animal model. Moreover, a significant decrease in pro-inflammatory bacteria was observed, as well as a substantial increase in beneficial populations such as *Bifidobacterium* [131]. Similarly, ginsenoside-Rb3 and ginsenoside-Rd effectively reduced the size and the number of the polyps and reinstated the dysbiotic gut microbial composition and intestinal microenvironment in Apc^Min/+^ mice by promoting the growth of SCFA-producing bacteria [132]. Given the encouraging results obtained from studies conducted both in vitro and in vivo, prebiotic administration has also been evaluated in more structured, randomized clinical trials involving CRC patients. A randomized, double-blind, no-treatment parallel control clinical trial involving 140 perioperative CRC patients was performed to investigate the effects of prebiotics containing fructooligosaccharides, xylooligosaccharides, polydextrose, and resistant dextrin on the immune system and the intestinal microbiota. In the preoperative interval, prebiotics upregulated serum levels of IgG, IgM, and transferrin, while in the postoperative period, they enhanced levels of IgG, IgA, CD8+ T cells, and total B lymphocytes. Prebiotic administration increased the abundance of *Bifidobacterium* and *Enterococcus* but decreased the abundance of *Bacteroides* in the preoperative timeframe. On the other hand, in the postoperative period, the abundance of *Bacteroides* was decreased, while *Escherichia-Shigella* was increased, suggesting that prebiotic intake is recommended to improve serum immunologic indicators in patients with CRC 7 days before operation, since surgical trauma can alter the gut microbiome [133]. However, recent studies have demonstrated that prebiotic interventions may exert variable effects in different individuals, probably due to differences in the host genetic background, which may plausibly explain the different tumor phenotype, oncogenic pathways, and, subsequently, the response to a specific intervention [134].

### 5.5. Postbiotics

Postbiotics are chemical compounds of microbial origin including short chain fatty acids, enzymes, peptides, teichoic acids, peptidoglycan-derived muropeptides, endo- and exo-polysaccharides, cell surface proteins, vitamins, plasmalogens, and organic acids [140]. The chemopreventive effects of acetate, butyrate, and propionate mixture were evaluated in AOM/DSS-treated mice, determining a significant reduction in tumor incidence and size. Moreover, SCFAs suppressed pro-inflammatory cytokine expression, including that of IL-6, TNF-α, and IL-17, as well as COX-2 and NF-κB [141]. The prophylactic effects of postbiotics were also shown by the oral intake of mitochonic acid 35, an indole compound, which ameliorated the disease activity index score and survival rate, reducing the macroscopic formation of colonic tumors in murine models of CRC. In addition, it was able to downregulate colonic TNF-α, IL-6, TGF-β1, and fibronectin 1 expression, suggesting its ability to inhibit CRC carcinogenesis [142]. Among the non-viable microbial cells, researchers have exploited the possible therapeutic effects of *Enterococcus faecalis.* In this regard, it was observed that pre-treatment of THP-1-derived macrophages with heat-killed *E. faecalis* inhibited NLRP3 inflammasome activation in response to fecal content or commensal microbes, *Proteus mirabilis* or *E. coli*, according to the reduction in casp-1 activation and IL-1β maturation. Moreover, experiments in vivo showed that *E. faecalis* improved the severity of intestinal inflammation, protecting mice from the formation of colon tumors [143]. Interestingly, postbiotics have also been evaluated as adjuvants of anti-cancer therapies. The combined efficacy of *L. acidophilus* cell lysates with an anti-CTLA-4 monoclonal antibody was tested in vivo. In contrast to anti-CTLA-4 monotherapy, *L. acidophilus* lysates attenuated body weight loss and the combined administration significantly protected mice against CRC development, suggesting an enhancement of anti-CTLA-4 antitumor activity. Moreover, the synergistic combination led to an increase in CD8+T cells, especially the effector memory T cells, a decrease in Tregs, and it alternatively activated macrophages (M2) in the TME. Additionally, pre-treatment with *L. acidophilus* lysate in vitro showed an immunomodulatory effect through the inhibition of M2 polarization and of IL-10 production by lipopolysaccharide-activated macrophages. Lastly, the combined administration significantly inhibited the abnormal increase in the relative abundance of *Proteobacteria* and partly counterbalanced CRC-induced dysbiosis in mice. Thus, CTLA-4 blocking antibodies in combination with the present lysates may be of importance for the development of new therapeutic strategies against CRC to be tested in clinical trials [144]. The postbiotic field is as yet a highly unknown area, considering that the number and diversity of bacterial metabolites are vast. Thus, their safety profile is still under preclinical and clinical evaluation [62].

### 5.6. FMT

Fecal microbiota transplantation (FMT)—i.e., the transfer of a microbial ecology from a healthy donor into a patient—is currently being explored as a therapeutic strategy to restore normobiosis, the normal state of the human intestinal microbiota, in different pathological contexts [145]. Since CRC is characterized by a status of dysbiosis, FMT is considered as a potential clinical application in patients. To date, FMT has only proved to be highly successful in treating recurrent and antibiotic refractory *Clostridiodes difficile* (*C. difficile*) infection, with cure rates of approximately 90% [129]. In a CRC mouse model, FMT was able to normalize the gut microbiota through the reduction of tumor growth. FMT contributed to reducing the levels of inflammation by decreasing IL-1β, IL-6, and TNF-α levels and increasing anti-inflammatory cytokines such as IL-10 and TGF-β through the inhibition of canonical NF-κB activity and cellular proliferation. Moreover, FMT treatment triggered the accumulation of Tregs but not Th1, Th2, and Th17 cells [146]. Additionally, the chemopreventive potential of FMT on FOLFOX-induced mucosal injury was evaluated. Microbiota transplantation reduced the severity of diarrhea and intestinal mucositis, suppressed IL-6 levels, and restored the number of goblet cells, zonula occludens-1, apoptotic, and NF-κB-positive cells, as well as the expression of toll-like receptors and MYD88. All these beneficial effects were accompanied by a restoration of the gut microbiota composition without causing bacteremia [147]. However, it is worth noting that the impact of FMT on the recipient immune system is complicated and unpredictable, and the risk of dissemination of unknown pathogens cannot be prevented. In addition, numerous questions remain to be answered, including the features that a “good donor” should present, the routes of administration, and the long-term effects of this therapy [148].

## 6. Conclusions and Perspective

The core of CRC carcinogenesis is also defined by gut microbiota metabolic activity and a dysbiotic composition. Hence, a consortium of inflammatory responses, virulence factors, and impaired epithelial signaling create a suitable microenvironment for the development of disrupted and irregular interactions between the host and the gut microbiota [149]. Even if surgery is the primary therapeutic option, patients with advanced disease or cancer recurrence after surgery remain difficult to cure. Since the gut microbiota is gaining more attention, a deeper knowledge of its interaction with the host’s immune system will elucidate the outcomes of cancer therapeutic strategies. Lastly, research is currently assessing the impact of personalized diets and biomodulators to restore a eubiotic condition for the prevention and treatment of CRC [150,151].

## Figures and Tables

**Figure 1 ijms-21-05389-f001:**
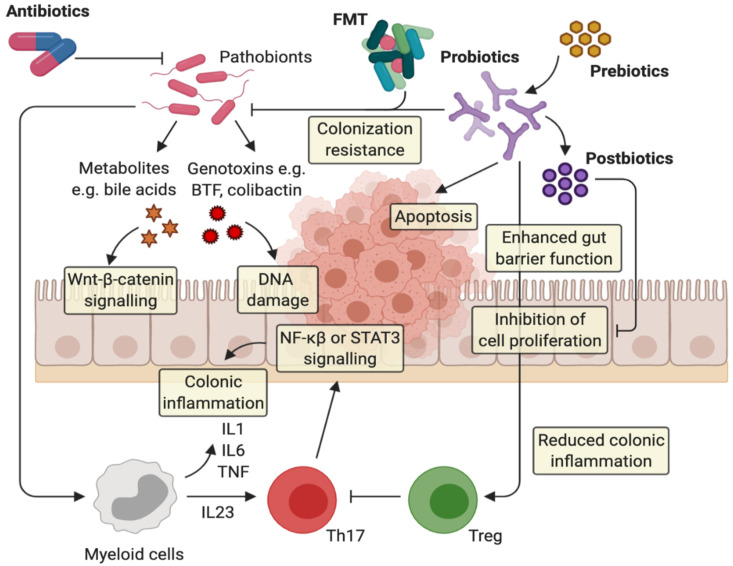
The gut microbiota can influence colorectal carcinogenesis via a variety of mechanisms, including microbial-derived factors such as metabolites or genotoxins. Skewed host–microbe interactions contribute to the activation of pro-carcinogenic inflammatory pathways that ultimately lead to the progression of CRC. Antibiotics usage is effective in eradicating pathobionts, but its non-selective antimicrobial actions can affect gut homeostasis by also killing health-promoting bacteria and, therefore, reducing its application in CRC management. Prebiotic function fosters probiotic growth. Probiotics act through different anticancerogenic mechanisms: (i) probiotics can inhibit the colonization of pathogenic bacteria, (ii) they can enhance barrier function increasing mucin production and tight junction protein expression, (iii) they promote homeostatic immune responses, contributing to the expansion of anti-inflammatory responses by Treg cells and the modulation of pro-inflammatory cytokine release, (iv) they promote apoptosis on cancer cells. Postbiotics induce selective cytotoxicity against tumor cells as well as the control of tumor cell proliferation by inhibiting NFATc3 activation. Finally, fecal microbiota transplantation (FMT) could be used in CRC management to restore microbiome normobiosis and therefore induce homeostatic immune responses; nevertheless, potential complications associated with FMT include the risk of introducing new pathobionts and the spreading of disease-associated genes.

**Table 2 ijms-21-05389-t002:** Examples of some commercially available probiotics.

Brand Name	Strain	Producer
Dicoflor	*Lactobacillus rhamnosus GG*	AGPHARMA
Enterogermina	*Bacillus clausii*	SANOFI
Enterolactis	*Lactobacillus casei*	SOFAR
Nutriflor	*Lactobacillus acidophilus DDS-1* *Lactobacillus bulgaricus DDS-14* *Bifidobacterium bifidum* *Lactobacillus rhamnosus*	NUTRIGEA
Probactiol Duo	*Lactobacillus acidophilus NCFM* *Lactobacillus paracasei Lpc-37* *Bifidobacterium lactis Bi-07* *Bifidobacterium lactis Bi-04*	METAGENETICS
VSL#3	*Streptococcus thermophilus* *Bifidobacterium breve* *Bifidobacterium longum* *Bifidobacterium infantis* *Lactobacillus acidophilus* *Lactobacillus plantarum* *Lactobacillus paracasei* *Lactobacillus delbrueckii subsp. bulgaricus*	FERRING FARMACEUTICI
Yakult	*Lactobacillus casei Shirota*	YAKULT (Tokio)

**Table 3 ijms-21-05389-t003:** Prebiotic-rich foods and their effects on human health.

Prebiotic	Origin	Clinical Benefit	References
Fructo-Oligosaccharides (FOS)	Vegetables, cereals (onion, garlic, artichokes)	Crohn’s disease Colitis CRC Obesity	[135]
Gluco-Oligosaccharides (GOS)	Legumes (lentils, chickpeas and broad beans)	Crohn’s disease Colitis Obesity	[135,136]
Ginsenoside-Rb3	Panax Ginseng	Myeloid leukemia CRC Heart failure	[132,137]
Inulin	Asparagus and artichokes	Crohn’s disease Colitis CRC Obesity Diabetes	[138]
Lactulose	Boiled milk	Constipation	[139]

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
