# Peer review of "Gut Microbiota Manipulation as a Tool for Colorectal Cancer Management: Recent Advances in Its Use for Therapeutic Purposes"

_ijms, 2020, doi:10.3390/ijms21155389_

Round 1

Reviewer 1 Report

The microbiota has become an important clinical target for CRC treatment. The manuscript by Perillo et al. compiles and analyses extensive literature data regarding gut microbiota in normal individuals, and patients suffering CRC, then reviews microbiota modulation as a therapeutic approach for CRC.

This manuscript is a comprehensive, well organized, complete, and accessible review. The less experienced reader will find the necessary information to catch up with the field, while the more experienced researcher will find useful analysis by the authors.

1) Except for a few sentences, the manuscript is understandable. However, I will strongly suggest the authors checking for language. Please give particular attention to style, usage of comma/ period/ hyphen, adj/adv form, verb agreement, and typographic mistakes. (A few examples are provided at the very end of this document, but I am not able to provide a complete list).
Please, consolidate and check the references. (Pages and volumes are frequently missing and there are typos in the titles).

2) The authors are using a lot of jargon that might be difficult for newcomers in the field (i.e.: Pathobionts, genotoxin, exposomal, mucositis, normobiosis, dysbiosis,…)

Please consider adding the definition of these terms in the text, or in a table if you find it more convenient. If you choose a table you might add other terms such: fecal microbiota transplantation, APCmin/+ mouse, etc...

Besides, the manuscript is full of acronyms and abbreviations. An initial section “Abbreviations” might be very useful. I also recommend the authors removing abbreviation used only once or twice, then prefer their full denomination, it may remove a burden to the reader: DSBs SBS ID BFT MUC sIgA NTBF DCs NK HDAC, IRI, OX, CAP XEL SPF, PUFAs, AP-1, LAB, GOS-lu, RCTs, MA-35, Fn1, M2, LPS, CDI, ZO-1, TLRs, etc… But, feel free to keep those more speaking to readers: for instance (Reg3γ, NFAT3c, etc…).On the other hand, please give the meaning for AOM/DSS, FOLFOX.

Abstract

The clarity and style of the two following sentences could be improved:

3) “Therefore, the gut microbiota may be a key therapeutic target for the management of CRC also in light of recent experimental data suggesting the role of the gut microbiota as modulator of response to cancer immunotherapies”

 4) “In this review, we will discuss recent advancements in the understanding of host-microbe interactions in CRC and the basis to promote homeostatic immune responses through microbe-targeted therapies useful in preventing/treating CRC development.”

Background

5) “The role of the gut microbiota in cancer biology has been increasingly recognized as an environmental factor favouring CRC development. Indeed, the gut microenvironment harbours a complex microbial ecosystem comprising approximately 3×1013 bacteria [11],”

Newcomers in the field might be confused that microbiota is only bacteria. Maybe authors could add somewhere the definition for microbiota: bacteria, fungus, virus, protist, etc...

Role of Gut microbiota in CRC

6) This paragraph is informative and focuses first on pro-oncogenic then anti-oncogenic effects of bacterial strains. May the authors consider separating the two using subparagraph titles?

7) I would like to suggest the option to add a table summarizing compiled data about bacteria: (genera, strain, pro-oncogenic /anti-oncogenic effect/ mechanism, etc…)?
If the authors choose this option, please consider adding information regarding bacteria provided later in the text.

8) In the same paragraph, APC and apc(min/+ mice) are using slightly similar acronym; it might be confusing.  Please consider giving a short explanation about what is apc(min/+) mouse for less experienced readers.

9) The following sentence should be toned:

 “Furthermore, CRC progression is affected by alteration of bacterial metabolism leading to production of oxidative stress molecules [43] [44] [29], promoting chronic inflammation and disrupting intestinal barrier integrity [35] [45]. Moreover, choline degradation by gut bacteria contributes to colon cancer via synthesis of trimethylamine N-oxide, a potential carcinogen [27].”

The cited papers rather suggest a relationship than bring a definitive demonstration.

Lifestyle, diet and microbiota in the early onset of CRC

9) “It is now widely recognised that the adoption of a westernized diet rich in red meat, saturated fat and low in fibres exerts a negative effect on intestinal homeostasis [58], promoting also gut dysbiosis and inflammation [59] [49].”
Please add more references to support the claim: “widely recognized”

10) “The demonstration of the increased risk of developing CRC due to westernized diet came from a study performing a two-week diet exchange in people from US with a high fat and lower fibre diet, and rural Africans, whose diet is characterized by a high-fibre and low-fat content.”

Please tone this sentence: it refers to a 2-weeks food exchange study that follows biomarkers for cancer risk. It shows an increase of biomarkers related to cancer risk, but it does not demonstrate directly an increased risk of developing CRC.

11) “Not only changes in the diet but also in agricultural practices and food additives (used to extend the shelf-life of processed foods) have“ and the following are difficult to follow. It will be nice to be more specific about agriculture practices and food additives to link with the following sentences concerning nitrite (fertilizers?), then synthetic dyes, monosodium glutamine (flavor enhancers) and titanium oxide (shiny appearance?)

Current treatments for CRC and role of the gut microbiota in their therapeutic efficacy

12) The paragraph title could definitively be improved for clarity.

13) Please indicate that XELODA is a brand name for capecitabine.

14) “However, a significant increase of some conditional pathogens is observed [7]”
Being more specific might help the reader to understand this sentence.

15) Does CAPeOX mean capecitabine and oxaliplatin?

16) “In addition, other gut microbes might aggravate chemotherapy-related adverse reactions via microbial metabolism of chemotherapy agents [76]. For instance, IRI effectiveness is severely limited by gastrointestinal tract toxicity caused by gut bacterial β-glucuronidase (GUS) enzymes [76].”

Add references or modify the sentence.

17) “Targeted therapies include three main groups of drugs: monoclonal antibodies against epidermal growth factor receptor (EGFR) (cetuximab and panitumumab), monoclonal antibodies against vascular endothelial growth factor (VEGF)-A (bevacizumab), fusion proteins that target multiple proangiogenic growth factors (e.g. aflibercept) and small-molecule-based multikinase inhibitors (e.g. regorafenib) [69]”.

It sounds like 4 drugs acting on 4 different targets, but 3 different types of molecules. Please consider modifying “include three main groups of drugs”.

 Clinical applications of the gut microbiota modulation for CRC prevention and management

Dietary intervention:

18) May the authors clarify regarding Akkermansia muciniphila?

“Vitamin D has also a significant regulatory effect on the homeostasis of the microbiota, especially on the regulation of the intestinal barrier integrity mediated by Akkermansia muciniphila, a mucin-degrading bacterium [89] [90]”(….) “In particular, food withdrawal decreased the abundance of potentially pathogenic Proteobacteria, while elevated A. muciniphila in mice fed with HFD. [93].”

Since the authors wrote “by Akkermansia muciniphila, a mucin-degrading bacterium”, it may appear in contradiction with figure legend:

“ii) enhance barrier function increasing mucin production and tight junction protein expression,”

 Antibiotics

19) Please consider rewriting the following sentence for clarity:

“BBR not only was able to reverse the microbiota imbalance induced by Fn, but also blocked the secretion of mucosal immune factors, such as IL-21, IL-22, IL-31 and CD40L, the activation of the Janus Kinase/Signal Transducer and Activator of Transcription (JAK/STAT) and mitogen-activated protein kinase/extracellular signal Regulated kinases (MAPK/ERK) pathway, which were stimulated by Fn in vivo [104].

 Probiotics

20) “In addition, this probiotic showed not only immunomodulatory effects by downregulating colonic IL-22, but also apoptotic effects by upregulating caspase (casp)-7, casp-9, and Bik expression [113].”

What about to be a more specific “apoptotic effect on cancer cells”?

Conclusion and perspectives

The two following sentences are hard to understand, please consider rewriting them:

21) ”The metabolic activity and composition of the gut microbiota defines the core of carcinogenesis in CRC in relation to intestinal dysbiosis.”

22) ”Understanding the immune and inflammatory mechanisms of microbiota mediate CRC pathogenesis, the interaction of the gut microbiota with the diet, lifestyles, drugs use and host’s metabolic parameters will help to explore the manipulation of the microbiota as an effective therapeutic strategy for the prevention and treatment of CRC through personalized gut microbiota biomodulators [134] [135].”

Figure 1

23) Replace toxins with genotoxins.

24) Action of probiotics on apoptosis is shown but not indicated in the legend.

--------------------------------------------------------------------

Example of sentences to be checked for language:

Adj/adverb form:
“in specific bacterial taxa with a potential detrimental impact on mucosal immune responses [24].” potentially?

Verb agreement:
“The development and progression of CRC is associated with the progressive failure of immunosurveillance”  are?
“A large body of literature have reported that faecal”  has?
“which the number of intestinal polyps were reduced as well” was?
“The chemopreventive effects of acetate, butyrate and propionate mixture was evaluated “ Were?
“considering that the number and diversity of bacterial metabolites is huge”. Are?

Typo
“which have been associated to a great increase” associated with?
“Modulation of the gut microbiota though the use of antibiotics was partially evaluated in CRC, with only few studies present in literature [98].” through?

Author Response

Response to Reviewer 1

This manuscript is a comprehensive, well organized, complete, and accessible review. The less experienced reader will find the necessary information to catch up with the field, while the more experienced researcher will find useful analysis by the authors.

We thank the referee for the careful and positive evaluation of our work.

Except for a few sentences, the manuscript is understandable. However, I will strongly suggest the authors checking for language. Please give particular attention to style, usage of comma/ period/ hyphen, adj/adv form, verb agreement, and typographic mistakes.

 We have checked the manuscript for the language and we corrected the typos and errors.

Please, consolidate and check the references.

 We updated the all the references along the manuscript

The authors are using a lot of jargon that might be difficult for newcomers in the field (i.e.: Pathobionts, genotoxin, exposomal, mucositis, normobiosis, dysbiosis,…)

 We provided along the manuscript the definition for the terms that could be difficult to understood for newcomers in the field, e.g. Pathobionts (i.e. microorganisms commonly living in the gut that become harmful under certain circumstances)

Besides, the manuscript is full of acronyms and abbreviations. An initial section “Abbreviations” might be very useful. I also recommend the authors removing abbreviation used only once or twice.

We provided a list for the abbreviations used in the manuscript

The clarity and style of the two following sentences could be improved:

3) “Therefore, the gut microbiota may be a key therapeutic target for the management of CRC also in light of recent experimental data suggesting the role of the gut microbiota as modulator of response to cancer immunotherapies”

 4) “In this review, we will discuss recent advancements in the understanding of host-microbe interactions in CRC and the basis to promote homeostatic immune responses through microbe-targeted therapies useful in preventing/treating CRC development.”

We rephrased the two sentences to be more compelling and understandable

5) “The role of the gut microbiota in cancer biology has been increasingly recognized as an environmental factor favouring CRC development. Indeed, the gut microenvironment harbours a complex microbial ecosystem comprising approximately 3×101 3 bacteria [11],”

Newcomers in the field might be confused that microbiota is only bacteria. Maybe authors could add somewhere the definition for microbiota: bacteria, fungus, virus, protist, etc...

We agree with the reviewer and we now indicate in line 60 a more compelling definition of the microbial composition of the gut microbiota

6) This paragraph is informative and focuses first on pro-oncogenic then anti-oncogenic effects of bacterial strains. May the authors consider separating the two using subparagraph titles?

According to the reviewer’s comment, we divided this section into two separate sub-paragraphs

7) I would like to suggest the option to add a table summarizing compiled data about bacteria: (genera, strain, pro-oncogenic /anti-oncogenic effect/ mechanism, etc…)?

If the authors choose this option, please consider adding information regarding bacteria provided later in the text.

We thank the review for this comment and accordingly we added a table (table 1) summarizing the potential role and mechanism of action of different gut microbes in CRC

8) In the same paragraph, APC and apc(min/+ mice) are using slightly similar acronym; it might be confusing.  Please consider giving a short explanation about what is apc(min/+) mouse for less experienced readers.

In line 109 we now briefly explain the phenotype of APCmin/+ mice

 The following sentence should be toned: “Furthermore, CRC progression is affected by alteratio n of bacterial metabolism leading to production of oxidative stress molecules [43] [44] [29], promoting chronic inflammation and disrupting intestinal barrier integrity [35] [45]. Moreover, choline degradation by gut bacteria contributes to colon cancer via synthesis of trimethylamine N-oxide, a potential carcinogen [27].”

The cited papers rather suggest a relationship than bring a definitive demonstration

According to the reviewer’s comment, we gave a more precautionary description on the role of the contribution of gut microbiota in CRC according to the cited literature

9) “It is now widely recognised that the adoption of a westernized diet rich in red meat, saturated fat and low in fibres exerts a negative effect on intestinal homeostasis [58], promoting also gut dysbiosis and inflammation [59] [49].”

Please add more references to support the claim: “widely recognized”

 We added more references to support our claim

10) “The demonstration of the increased risk of developing CRC due to westernized diet came from a study performing a two-week diet exchange in people from US with a high fat and lower fibre diet, and rural Africans, whose diet is characterized by a high-fibre and low-fat content.”

Please tone this sentence: it refers to a 2-weeks food exchange study that follows biomarkers for cancer risk. It shows an increase of biomarkers related to cancer risk, but it does not demonstrate directly an increased risk of developing CRC.

We thank the reviewer for this comment and we rephrased the sentence to be more compelling

11) “Not only changes in the diet but also in agricultural practices and food additives (used to extend the shelf-life of processed foods) have“ and the following are difficult to follow. It will be nice to be more specific about agriculture practices and food additives to link with the following sentences concerning nitrite (fertilizers?), then synthetic dyes, monosodium glutamine (flavor enhancers) and titanium oxide (shiny appearance?)

We added few more examples to better explain the link between the gut microbiota, new food practices and CRC development from line 183 to 201.

12) The paragraph title could definitively be improved for clarity.

We thank the reviewer for this comment and we modified the paragraph’s title in “Relevance of the gut microbiota in the efficiency of cancer therapies”

13) Please indicate that XELODA is a brand name for capecitabine.

We indicated the brand name according to this comment

14) “However, a significant increase of some conditional pathogens is observed [7]”

Being more specific might help the reader to understand this sentence.

In line 216 we added the species associated with these alterations in the microbiota

15) Does CAPeOX mean capecitabine and oxaliplatin?

Yes, CAPeOX mean capecitabine and oxaliplatin. We reported the definition of CAPeOX in the abbreviation table.

16) “In addition, other gut microbes might aggravate chemotherapy-related adverse reactions via microbial metabolism of chemotherapy agents [76]. For instance, IRI effectiveness is severely limited by gastrointestinal tract toxicity caused by gut bacterial β-glucuronidase (GUS) enzymes [76].”

Add references or modify the sentence.

We added two references to support this claim

17) “Targeted therapies include three main groups of drugs: monoclonal antibodies against epidermal growth factor receptor (EGFR) (cetuximab and panitumumab), monoclonal antibodies against vascular endothelial growth factor (VEGF)-A (bevacizumab), fusion proteins that target multiple proangiogenic growth factors (e.g. aflibercept) and small-molecule-based multikinase inhibitors (e.g. regorafenib) [69]”.

It sounds like 4 drugs acting on 4 different targets, but 3 different types of molecules. Please consider modifying “include three main groups of drugs”.

We modified according to this comment in line 234

19) Please consider rewriting the following sentence for clarity:

“BBR not only was able to reverse the microbiota imbalance induced by Fn, but also blocked the secretion of mucosal immune factors, such as IL-21, IL-22, IL-31 and CD40L, the activation of the Janus Kinase/Signal Transducer and Activator of Transcription (JAK/STAT) and mitogen-activated protein kinase/extracellular signal Regulated kinases (MAPK/ERK) pathway, which were stimulated by Fn in vivo [104].

We rephrased the sentence in lines 323-325 to be more compelling

20) “In addition, this probiotic showed not only immunomodulatory effects by downregulating colonic IL-22, but also apoptotic effects by upregulating caspase (casp)-7, casp-9, and Bik expression [113].”

What about to be a more specific “apoptotic effect on cancer cells”?

We modified the sentence in line 350 by indicating the mediators of the apoptotic effect, i.e. “caspase (casp)-7, casp-9 and Bik, as well as a decrease in Ki67 expression”

The two following sentences are hard to understand, please consider rewriting them:

21) ”The metabolic  activity and composition of the gut microbiota defines the core of carcinogenesis in CRC in relation to intestinal dysbiosis.”

22) ”Understanding the  immune and inflammatory mechanisms of microbiota mediate CRC pathogenesis, the interaction of the gut microbiota with the diet, lifestyles, drugs use and host’s metabolic parameters will help to explore the manipulation of the microbiota as an effective therapeutic strategy for the prevention and treatment of CRC through personalized gut microbiota biomodulators [134] [135].”

We rephrased the sentences to improve the clarity of this section

Figure 1

23) Replace toxins with genotoxins.

24) Action of probiotics on apoptosis is shown but not indicated in the legend.

We modified figure 1 according to this comment

Reviewer 2 Report

The present review by Perillo F. et al, titled "Gut microbiota manipulation as a tool for colorectal cancer management : recent advances on its use for therapeutic purposes" a well written and complied by authors and consistently presented. However I would like to some missing points, Add some commercially available probiotics and prebiotics in the review in table formate. 

Author Response

Response to Reviewer 2

The present review by Perillo F. et al, titled "Gut microbiota manipulation as a tool for colorectal cancer management: recent advances on its use for therapeutic purposes" a well written and complied by authors and consistently presented. However I would like to some missing points, Add some commercially available probiotics and prebiotics in the review in table formate

We thank the reviewer for the careful analysis of our paper and according to his/her comment we added two tables (table 2 and 3) reporting the commercially available pro- and prebiotics.

Reviewer 3 Report

The authors present a review about microbiota in colorectal cancer. There are some issues to be addressed:

The manuscript needs a deep English review

Paper format and references do not follow IJMS standards

Some imperfections throughout the manuscript such as:

  • "Over the past three decades, molecular genetic studies have revealed..." the sentences is not complete. There was described some other pathways (methylator phenotype pathway, Microsatellite instability pathway..) the authors should complete the sentence.
  • " The development and progression of CRC is associated with the
    progressive failure of immunosurveillance..." The sentence is sharping. This is not the only way to CRC progression..
  • "Currently, checkpoint inhibitors have been
    investigated with promising responses" It is not correct. IO now is an standard of care in patients with CRC MSI

I suggest authors to include a graphical abstract

Author Response

Response to Reviewer 3

The manuscript needs a deep English review

We corrected the manuscript for typos and errors

Paper format and references do not follow IJMS standards

We modified the references to comply with the IJMS standards

Some imperfections throughout the manuscript such as:

  • "Over the past three decades, molecular genetic studies have revealed..." the sentences is not complete. There was described some other pathways (methylator phenotype pathway, Microsatellite instability pathway..) the authors should complete the sentence.

We modified this sentence to be more compelling.

  • " The development and progression of CRC is associated with the

progressive failure of immunosurveillance..." The sentence is sharping. This is not the only way to CRC progression.

We agree with the reviewer and in line 53 we now account for the multifactorial nature of CRC progression

  • "Currently, checkpoint inhibitors have been investigated with promising responses" It is not correct. IO now is an standard of care in patients with CRC MSI

We agree with the reviewer and now we state in line 244 that IO is a standard of care in patients with CRC MSI

I suggest authors to include a graphical abstract

We added a graphical abstract at the bottom of the manuscript
